# Structural Responses Estimation of Cable-Stayed Bridge from Limited Number of Multi-Response Data

**DOI:** 10.3390/s22103745

**Published:** 2022-05-14

**Authors:** Namju Byun, Jeonghwa Lee, Joo-Young Won, Young-Jong Kang

**Affiliations:** 1Future and Fusion Laboratory of Architectural, Civil and Environmental Engineering, Korea University, Seoul 02841, Korea; skawn0702@naver.com (N.B.); qevno@korea.ac.kr (J.L.); 2School of Civil, Environmental and Architectural Engineering, Korea University, Seoul 02841, Korea; jywon27@korea.ac.kr

**Keywords:** cable-stayed bridge, multi-response data, deformed shape, member internal force, SRALMR

## Abstract

A cable-stayed bridge is widely adopted to construct long-span bridges. The deformation of cable-stayed bridges is relatively larger than that of conventional bridges, such as beam and truss types. Therefore, studies regarding the monitoring systems for cable-stayed bridges have been conducted to evaluate the performance of bridges based on measurement data. However, most studies required sufficient measurement data for evaluation and just focused on the local response estimation. To overcome these limitations, Structural Responses Analysis using a Limited amount of Multi-Response data (SRALMR) was recently proposed and validated with the beam and truss model that has a simple structural behavior. In this research, the structural responses of a cable-stayed bridge were analyzed using SRALMR. The deformed shape and member internal forces were estimated using a limited amount of displacement, slope, and strain data. Target structural responses were determined by applying four load cases to the numerical model. In addition, pre-analysis for initial shape analysis was conducted to determine the initial equilibrium state, minimizing the deformation under dead loads. Finally, the performance of SRALMR for cable-stayed bridges was analyzed according to the combination and number of response data.

## 1. Introduction

The construction of bridges with longer spans is a challenging task in civil engineering. A cable-stayed bridge is widely adopted to construct long-span bridges, in which the cable connects the bridge deck and the pylon to support the bridge deck. The forces generated in cables can be divided into horizontal and vertical components. The horizontal component of the cable force causes a significant compressive force on the girder, and the girder has a complex distribution of bending moment owing to the vertical component of the cable force. In addition, horizontal and vertical components of the cable force cause a bending moment and compressive force on the pylon, respectively. The brokenness of one cable can affect the entire equilibrium system of the structure. Therefore, ensuring the integrity of cable-stayed bridges is a significant factor in structural maintenance. Various responses and environmental conditions have been measured using a monitoring system to evaluate the integrity of cable-stayed bridges [1,2,3].

If the Structural Health Monitoring (SHM) that can evaluate the integrity of the structure using measured data such as displacement, strain, slope, acceleration, and temperature is successfully applied to on-site structure, the limitations of manpower and budget for maintenance can be overcome. The main objective of SHM is the detection of degradation of structure and to provide useful data for maintenance decision making. The research field for SHM can be divided into three fields according to the objective: damage detection [4,5], response pattern prediction [6], and unmeasured response estimation (URE) [7,8,9]. This paper is a study on the field of URE. In previous studies for URE, measured data (displacement, slope, strain, and acceleration) related to the structural stiffness has been generally utilized.

Displacement, which is a structural response, is typically utilized to indirectly evaluate the performance of a structure. The performance of the structure was indirectly evaluated by comparing the measured displacement with the deformation limit determined via numerical analysis and a loading test. Linear variable differential transformers (LADTs) and laser Doppler vibrometers (LDVs) are widely used to measure the displacement of structures. However, these conventional displacement sensors have limitations, such as the requirement of additional fixed reference points and high sensitivity to the surrounding climate. To overcome these limitations, studies have been conducted on displacement estimation methods using other response data.

Among the analyzed structural responses, acceleration, strain, and slope data have been widely used to estimate the displacement of structures. Using acceleration data, displacement can be estimated based on the relation that displacement is a double integration of acceleration. Park et al. [10] estimated the displacement through double integration of acceleration, assuming that the initial displacement and average velocity were zero. Lee et al. [11] proposed a time-domain finite impulse response filter (FIR filter) for estimating displacement without initial conditions in a way that minimizes the square error between acceleration, which is assumed as the central finite difference of displacement, and measured acceleration. In addition, a wireless displacement measurement system using the FIR filter proposed by [11] was developed and validated with experimental acceleration data [12]. These methods use acceleration for accurate estimations at low prices; however, only the displacement at positions where the acceleration sensors are installed can be estimated because the spatial parameter is not included in the acceleration response.

In contrast, the slope and strain response, which include spatial parameters, can be used to estimate the displacement at positions where sensors are not installed. The shape superposition method (SSM) is mostly adopted for estimation methods that use slope and strain data. SSM consists of a shape function and weight factor. The weight factor is derived by minimizing the least-square error between the measured response and product of shape function and weight factor. Hou et al. [13] estimated displacement using slope data based on the SSM. The power series was utilized as a shape function, and the coefficient was considered to represent the boundary condition. The displacement estimation using strain data was firstly conducted by Foss and Haugse [14] based on the modal mapping. Modal mapping is SSM using the mode shape as a shape function. For general use of modal mapping, a theoretical mode shape composed of sine functions was adopted as the shape function, and its effectiveness was validated with experimental data from various structures [15]. A study using mode shape derived by frequency analysis of a finite element method (FEM) model was also conducted to improve the accuracy of displacement estimation [16]. The results of the research showed that the FEM mode shape can estimate the displacement more accurately than the theoretical mode shape. In addition, modal mapping proposed by [14] was widely adopted as a basic method to estimate displacement [17,18,19,20,21].

Recently, studies on displacement estimation using strain and acceleration data fusion have been conducted to cope with the drawback of estimation by one type of data. Park et al. [22] developed an indirect displacement estimation using acceleration and strain (IDEAS) method by combining FIR filter [11] and modal mapping [14]. The estimation by the IDEAS method showed better accuracy than that by only strain data. Then, the extendibility of the IDEAS method was investigated for various types of beam structures [23]. The Kalman filter, actively studied for the accuracy enhancement in aerospace engineering, was also applied to the IDEAS [24]. IDEAS using FIR and Kalman filter has been widely adopted to indirectly estimate displacement based on the strain and acceleration data [25,26,27].

Although the previously proposed methods can indirectly well estimate displacement, they use one or two types of response data and focus on the estimation of displacement at specific points. To accurately evaluate the integrity of the structure, the response shape for the entire structure is required, not the displacement at specific points. In addition, in order to apply the estimation method to the monitoring system in which the sensor is already installed, various combinations of response data need to be available to the algorithm. As the Global Navigation Satellite System (GNSS) has been developed, which overcomes the limitations of conventional displacement sensors, the “Structural Responses Analysis using a Limited amount of Displacement data” (SRALD) was proposed [28,29]. The deformed shape was estimated using the displacement data, and the internal force was determined using the deformed shape using the stiffness method. In addition, SRALD was validated for the beam, truss, and cable-stayed bridge finite element method (FEM) models by comparing the estimated responses with values estimated by the spline interpolation method. Subsequently, SRALD was improved to the “Structural Responses Analysis using a Limited amount of Multi-Response data” (SRALMR), which uses the slope and strain response as additional data to reduce the required number of expensive GNSS [30]. In addition, SRALMR was applied to the beam and truss FEM models, and its effectiveness was verified by comparing it with SRALD. The SRALMR was established by extending the shape superposition theory basically used in previous estimation studies to use the three response data in various combinations. Compared to the previously developed estimation methods, the major strength of the SRALMR is that it can utilize any combination of displacement, slope, and strain data and estimate the response shape of the entire structure rather than the response of a specific point. However, the validation of SRALMR for structures that have complex structural behavior compared to beams and trusses is required for on-site applications.

Therefore, the validation of SRALMR for a cable-stayed bridge FEM model that has complex structural behavior owing to the cable is conducted in this study. The deformed shape was estimated using a limited amount of displacement, slope, and strain response data. Additionally, the girder axial force, girder moment, and cable force were determined using the deformed shape estimated by using the stiffness method. The pylon is also an important element for the entire equilibrium system, but it has a simple distribution of internal force, and its integrity can be relatively simply estimated using displacement data at the top of the pylon. For this reason, internal force estimation in this study is limited to the cable and girder elements. For the deformed shape and internal force, the estimation performance of SRALMR for the cable-stayed bridge was analyzed according to the combination and number of response data.

## 2. Estimation Algorithm

Choi et al. [28] developed an SRALD technique that can estimate the deformed shape and internal force from limited displacement data. Byun et al. [30] introduced SRALMR by improving SRALD to use slope and strain data for estimation. SRALMR is based on the shape-superposition method for the structural shape of each response. In a linear problem, each response of the structure can be expressed as the product of the shape function Φ and the weight factor α, as shown in Equations (1)–(3). $\Phi_{u}$, $\Phi_{\theta }$, and $\Phi_{\epsilon }$, are the shape functions of the displacement, slope, and strain, respectively. The weight factors for displacement u, slope θ, and strain ϵ are the same because the slope and strain are the first and second derivatives of the displacement, respectively. This is the basic concept of SRALMR.
(1)$$\left\{u\right\}=\left[\Phi_{u}\right]\left\{\alpha \right\}$$
(2)$$\left\{\theta \right\}=\left[\Phi_{\theta }\right]\left\{\alpha \right\}={\left[\Phi_{u}\right]}^{\prime }\left\{\alpha \right\}$$
(3)$$\left\{\epsilon \right\}=\left[\Phi_{\epsilon }\right]\left\{\alpha \right\}={\left[\Phi_{u}\right]}^{\prime \prime }\left\{\alpha \right\}$$

In this study, the structural shape function (SSF) is derived by applying a unit load to each node of the numerical model. The *i*th SSF $\Phi_{i,u}$, $\Phi_{i,\theta }$, and $\Phi_{i,\epsilon }$ for displacement, slope, and strain are represented as Equations (4) and (6), respectively. Finally, comprehensive SSF $\Phi_{i}$ in Equation (7) can be derived by combining all the SSF values for each response in one matrix.
(4)$$\Phi_{i,u}{}^{T}=\left[u_{i1}\;u_{i2}\;u_{i3}\;u_{i4}\;\cdots \;u_{iN_{dof,u}}\right]$$
(5)$$\Phi_{i,\theta }{}^{T}=\left[\theta_{i1}\;\theta_{i2}\;\theta_{i3}\;\theta_{i4}\;\cdots \;\theta_{iN_{dof,\theta }}\right]$$
(6)$$\Phi_{i,\epsilon }{}^{T}=\left[\epsilon_{i1}\;\epsilon_{i2}\;\epsilon_{i3}\;\epsilon_{i4}\;\cdots \;\epsilon_{iN_{dof,\epsilon }}\right]$$
(7)$$\Phi_{i}=\left[\begin{array}{c} \Phi_{i,u}\\ \Phi_{i,\theta }\\ \Phi_{i,\epsilon } \end{array}\right]\left(N_{dof}\times 1\right)$$

Here, $N_{dof}=N_{dof,u}+N_{dof,\theta }+N_{dof,\epsilon }$ is the number of degrees of freedom (DOF), $N_{dof,u}$, $N_{dof,\theta }$, and $N_{dof,\epsilon }$ are the numbers of DOF for each response, respectively; and $u_{iN_{dof,u}}$, $\theta_{iN_{dof,\theta }}$, and $\epsilon_{iN_{dof,\epsilon }}$ are the ith displacement, slope, and strain value at each DOF, respectively.

As indicated above, the structural response shape can be represented as a superposition of the SSF. An arbitrary response shape $\tilde{ARS}$ needs to be defined to establish an error function with measurement data. As shown in Equation (8), $\tilde{ARS}$ is defined as the superposed SSF multiplied by the weight factor. $N_{sf}$ is the number of shape functions used to estimate the structural response, and the $\tilde{ARS}$ in matrix form is represented by Equation (9).
(8)$$\tilde{ARS}=\alpha_{1}\Phi_{1}+\alpha_{2}\Phi_{2}+\alpha_{3}\Phi_{3}+\cdots +\alpha_{N_{sf}}\Phi_{N_{sf}}=\overset{N_{sf}}{\underset{i=1}{\sum }}\alpha_{i}\Phi_{i}$$
(9)$$\tilde{ARS}=\left[\Phi_{1}{\;\Phi }_{2}{\;\Phi }_{3}\;\cdots {\;\Phi }_{N_{sf}}\right]\left[\begin{array}{c} \;\alpha_{1}\;\\ \;\alpha_{2}\;\\ \;\alpha_{3}\;\\ \vdots \\ \alpha_{N_{sf}} \end{array}\right]\left(N_{dof}\times 1\right)$$

$\tilde{ARS}$ is composed of response values at each DOF. The measurement data matrix *MD* is composed of the response values measured at certain points where the sensors are installed. ω, ∅, and ε are the measurement data values for displacement, slope, and strain, respectively. The *MDs* for each response are defined in Equations (10)–(12), and $N_{md}$ is the number of measurements. A comprehensive *MD* to establish the error function with $\tilde{ARS}$ is constructed by combining the *MD* for each response.
(10)$$MD_{\omega }{}^{T}=\left[\omega_{1}\;\omega_{2}\;\omega_{3}\;\omega_{4}\;\cdots \;\omega_{N_{md,\omega }}\right]$$
(11)$$MD_{\emptyset }{}^{T}=\left[\emptyset_{1}\;\emptyset_{2}\;\emptyset_{3}\;\emptyset_{4}\;\cdots \;\omega_{N_{md,\emptyset }}\right]$$
(12)$$MD_{\varepsilon }{}^{T}=\left[\varepsilon_{1}\;\varepsilon_{2}\;\varepsilon_{3}\;\varepsilon_{4}\;\cdots \;\omega_{N_{md,\varepsilon }}\right]$$
(13)$$MD=\left\{\begin{array}{c} MD_{\omega }\\ MD_{\emptyset }\\ MD_{\varepsilon } \end{array}\right\}\left(N_{md}\times 1\right)$$

The weight factor required to estimate the deformed shape can be calculated using *MD* and $\tilde{ARS}$. Owing to the difference in the matrix size between *MD* and $\tilde{ARS}$, the matrix size of the $\tilde{ARS}$ needs to be adjusted to establish the error function. The $\tilde{ARS}$ is converted to an *ARS* matrix including only response values for DOF where the measurement data exist. Finally, the error function *E* is defined as the sum of the square errors between *MD* and *ARS* in Equation (14). Partial differentiation is performed with respect to the weight factor to minimize the error, which must equal zero. The calculation process is represented by Equations (15)–(17).
(14)$$E=\overset{N_{md}}{\underset{j=1}{\sum }}{\left(MD_{j}-ARS_{j}\right)}^{2}$$
(15)$$\frac{\partial E}{\partial \alpha_{k}}=2\sum_{j=1}^{N_{md}}\left[\left(MD_{j}-ARS_{j}\right)\left(\frac{\partial MD_{j}}{\partial \alpha_{k}}-\frac{\partial ARS_{j}}{\partial \alpha_{k}}\right)\right]=0\;\mathrm{where}\;k=\;1,\;2,\;\cdots ,\;N_{sf}$$
(16)$$\frac{\partial E}{\partial \alpha_{k}}=\sum_{j=1}^{N_{md}}\left[\left(\sum_{i=1}^{N_{sf}}\alpha_{i}\Phi_{ij}\right)\left(\Phi_{kj}\right)\right]=\sum_{j=1}^{N_{md}}\left[\left(MD_{j}\right)\left(\Phi_{kj}\right)\right]\;\mathrm{where}\;k=\;1,\;2,\;\cdots ,\;N_{sf}$$
(17)$${\left[\Phi \right]}_{N_{md}\times N_{sf}}^{T}{\left[\Phi \right]}_{N_{md}\times N_{sf}}{\left\{\alpha \right\}}_{N_{sf}\times 1}={\left[\Phi \right]}_{N_{md}\times N_{sf}}^{T}{\left\{MD\right\}}_{N_{md}\times 1}$$

The weight factor that minimizes the square error between ASD and MD can be calculated by Equation (17), using the inverse matrix of ${\left[\Phi \right]}^{T}\left[\Phi \right]$. However, if the $N_{sf}$ is greater than the $N_{md}$, ${\left[\Phi \right]}^{T}\left[\Phi \right]$ becomes a rank-deficient matrix. In general, the amount of measurement data is limited owing to the cost. To solve this problem, the singular value decomposition (SVD) method is adopted to solve Equation (17). By substituting the weight factor calculated using the SVD method in Equation (1), the structural deformation shape can be estimated.

Based on the stiffness method, the internal force can be determined using the relative rotation and displacement of the nodes attached to the element. Therefore, the internal structural force can be estimated using the estimated deformed shape (EDS). In this study, the internal force of a cable-stayed bridge was estimated by applying EDS to the numerical model as the displacement load.

## 3. Validation Process

The most representative characteristic of a cable-stayed bridge, which differs from other types of bridges, is the cable connected to girders and pylons. The cable force supports the girder and can be decomposed into vertical and horizontal forces. The horizontal component of the cable force causes a compression force in the girder. However, the girder has a complex moment shape owing to the vertical component of the cable force. Therefore, cable-stayed bridges consist of geometric nonlinearity, such as the cable sag effect, beam–column effect, and large displacement, whereas material nonlinearity arises when bridge elements exceed their individual elastic limits. The nonlinear effects must be considered when analyzing the ultimate behavior of a cable-stayed bridge [31].

However, cable-stayed bridges under normal operating conditions can be analyzed using a linear model. Ren [32] investigated the ultimate behavior of cable-stayed bridges up to failure, considering the material and geometric nonlinearity. The investigation results demonstrated that the behavior of a cable-stayed bridge is affected by geometric nonlinearity when the live load is four times greater than that of the dead load, and it is affected by material nonlinearity when the live load is two times greater than that of the dead load. In general, the dead load of a bridge is larger than the live load. Namely, the nonlinearity of a cable-stayed bridge is sufficiently small to be ignored under normal operating conditions. Therefore, SRALMR based on the superposition method can be applied to cable-stayed bridges.

The validation process of SRALMR for cable-stayed bridges is illustrated in Figure 1. The analysis of cable-stayed bridges is divided into two parts: initial shape analysis considering geometric nonlinearity and linear analysis with live load. The initial shape analysis was performed to determine the optimal cable forces that ensured minimal deformation of the structure under dead load conditions; whereas the linear analysis with the live load was performed to represent the structural behavior generated by the live load. The results of the linear analysis with the live load are used to assume the real response of the structure because the sensors installed after construction are complete and can only measure the response data generated by the live load. In addition, the response data (displacement, strain, and slope) at a limited point from the real response of the structure are utilized as measurement data for deriving the error function. The deformed shape and internal force of the target model are determined by superposing the results of the initial shape analysis and linear analysis with a live load.

SRALMR is applied to estimate the deformed shape and internal force generated by a live load because the sensor usually measures the response data generated by the live load. Therefore, the SSF and ARS required for estimation are derived from a linear analysis with a live load. The error function is then established using the ARS and MD. The deformed shape and internal force were estimated using the calculated weight factor to minimize the error function. Finally, the SRALMR was validated by comparing the results of the estimated and target models.

## 4. Numerical Model for Validation

### 4.1. Validation Model

The radiating-type cable-stayed bridge shown in Figure 2, which has three spans, two pylons, and forty cables, was used as the validation model in this study. The total length of the bridge was 920 m, and the lengths of the three spans were 220, 480, and 220 m, respectively. The height of each pylon was 165 m. The girder was composed of 185 nodes and 184 beam elements, and each pylon had 34 nodes and 33 beam elements. However, a truss element with no compression was used for the cable. Both ends of the girder were in the roller-supported condition, and both bottoms of the pylon were in a fixed condition. At the connections between the girder and pylon, a roller for the girder and free for pylon were applied for the boundary condition. Table 1 presents the materials and geometric properties of the main members. Abaqus 2022 is used for numerical analysis of the validation model. Girder and pylon are modeled by the beam element, and the cable is modeled by a truss element with no compression. In addition, all analyses for the initial shape, structural shape function, and target model are material and geometrically linear.

### 4.2. Initial Shape Analysis

The deformation and internal force of a cable-stayed bridge are critically dependent on the cable force. Therefore, it is important to determine the cable force and initial equilibrium state to minimize the deformation under dead loads. Various methods of initial shape analysis, such as the trial-and-error approach, successive substitution method, and target configuration under dead load (TCUD), have been suggested [33,34,35]. In this study, the initial force method was used to determine the initial equilibrium state of the validation model. In the initial force method, the geometric nonlinear analysis is repeated considering the internal force of all members at the current iteration as an initial internal force at the next iteration. Figure 3 presents the displacement of the girder center and pylon top according to the number of iterations. The cable forces for the original model are zero, and the appropriate cable forces that satisfy the equilibrium state under a dead load are determined by iteration. As the iterations progressed, the displacement converged to zero. The displacements of iteration 6 were 3.86 cm for the girder center and 0.38 cm for the pylon top. This is negligibly small compared to the span length of 450 m. Therefore, the deformed shape and internal force at iteration 6 were used for the results of the initial shape analysis.

### 4.3. Target Model

In this study, the target models used for the validation of SRALMR were derived using the numerical model. Figure 4 presents the deformed shape of the target model according to the applied force, and the magnitudes of each force are presented in Table 2. The static load is applied for target models, and the magnitudes of each load are determined for sufficient variation of internal force compared to the initial internal force generated by self-weight. Only one concentrated load is applied to each span for TM1 and TM2. Different loads act on two points for TM3. TM4 is the most complex case, in which three different loads are applied. The displacement, slope, and strain data of each target model were used as the measurement data for the validation of SRALMR.

### 4.4. Structural Shape Function (SSF)

The SSF for the validation model was first derived to estimate the deformed shape and the internal force. The SSF can be derived by applying a unit load on each node of the girder, excluding the boundary conditions. The total number of SSFs was 181. As indicated above, each SSF is composed of displacement, slope, and strain response data. The horizontal and vertical directions were considered for the displacement data, whereas only the longitudinal direction was considered for the slope and strain data. In addition, the displacement and slope data were derived from the nodes, whereas the strain data were derived from the elements. The number of nodes and elements were 253 and 290, respectively. Therefore, the matrix size of the SSF established for the validation model is 1049 × 181.

### 4.5. Measurement Location

Each element of the response data of the target model at the limited points was used as measurement data for validation in this study. The location of the measurement data can significantly affect estimation accuracy. Therefore, an effective sensor placement method has been studied. Kammer [36] first introduced the effective independence (EI) method, which maximizes both the spatial independence and signal strength of the shape function by maximizing the determinant of the associated Fisher information matrix. Papadopoulos and Garcia [37] proposed a driving point residue (DPR) coefficient to overcome the drawback of the EI method, which can select a location with a low energy content. However, the EI and EI-DPR methods can only be applied when the number of shape functions is larger than the number of measurement data. Therefore, the EI-DPR-distance method proposed by Byun et al. [30] was adopted to estimate the deformed shape and internal force. Equations (18)–(20) express the EI-DPR-distance method. The distance coefficient $d_{i}$ is the minimum value among the distances from the *i*th candidate location to each previously selected sensor location. The *i*th candidate location with the highest effective independence distribution $E_{D_{i}}$ was selected as the sensor location, and the distance coefficient was recalculated by considering the previously selected sensor locations. This procedure was repeated until the selected number of locations for each response was equal to the planned number of sensors. The EI-DPR-distance method can overcome the limitations of the EI and EI-DPR methods and consider the effects of different response sensor locations.
(18)$$\left[E\right]=\left[\Phi \right]{\left({\left[\Phi \right]}^{T}\left[\Phi \right]\right)}^{-1}{\left[\Phi \right]}^{T}$$
(19)$$DPR_{i}=\overset{N_{sf}}{\underset{j=1}{\sum }}\frac{\Phi_{ij}^{2}}{{\left|\Phi_{j}\right|}_{max}}$$
(20)$$E_{D_{i}}=diagonal{\left(\left[E\right]\right)}_{i}\times DPR_{i}\times d_{i}$$

If the sensor location is determined via the EI-DPR-distance method, the sensor layout case is numerous according to the number of each response sensor due to the distance coefficient. For this reason, the only example of measurement location using one displacement sensor and seven displacement sensors is represented in Figure 5 and Figure 6, respectively. As shown in Figure 5a and Figure 6a, the measurement location of the slope is different due to the measurement location of the displacement. In addition, Figure 5b,c shows the different measurement locations of the strain due to the measurement location of the slope. All combinations of response data are used for deformed shape validation and only three combinations of response data are used for internal force validation.

## 5. Validation Results

In this study, the application of SRALMR was validated using a numerical model of a cable-stayed bridge. The deformed shape and internal forces (girder axial force, girder moment, and cable force) were estimated from limited multi-response data. The estimated response shape (ERS) was then compared with the target response shape (TRS). The accuracy of the estimation according to the number of measurement data was analyzed using the normalized mean absolute percent error (NMAPE) represented in Equation (21), where $N_{dof}$ is the number of DOFs.
(21)$$NMAPE\left(\%\right)=\frac{100}{N_{dof}}\overset{N_{dof}}{\underset{i=1}{\sum }}\frac{\left|TRS_{i}-ERS_{i}\right|}{{\left|TRS\right|}_{max}}$$

### 5.1. Deformed Shape

In this section, the deformed shapes for TM1-4 are estimated using the SSF and EI-DPR-distance methods to identify the internal force. The parameter for estimating the deformed shape was the number of measurement data points for each response. The number of measurement data varied from one to seven for the displacement and zero to six for the slope and strain. The number of slope and strain data points increases equally. Based on the EI-DPR-distance method, the position of the displacement sensors was first determined by considering the location of the boundary conditions, and the position of the slope sensors was determined according to the position of the displacement sensors. Finally, the strain sensors were placed considering the displacement and slope sensors.

Figure 7 presents the NMAPE of EDS according to the number of measurements. The NMAPE is calculated considering the vertical and horizontal deformed values of all the nodes. The maximum NMAPEs are 11.04% for TM1, 11.10% for TM2, 11.51% for TM3, and 13.51% for TM4. The difference in the maximum NMAPE for each TM is not large because the deformed shape of the pylon can be estimated using only one displacement data point on the girder center. The NMAPE of the EDS decreased as the number of measurement data points increased. The estimation results demonstrated that displacement data, including vertical and horizontal values, are the most effective estimation responses. However, slope and strain data, including only one-dimensional values, can also be used to enhance the estimation accuracy. Namely, slope and strain data can reduce the number of expensive GNSSs required to estimate the proper deformed shape. Figure 8 presents the position of the sensors for each response and the enhancement of the estimation accuracy by the slope and strain data. The target deformed shape (TDS), EDS with only one displacement data, and EDS with one displacement data, six slope data, and six strain data are compared in Figure 8.

### 5.2. Internal Force

The displacement value at a certain point of the structure has generally been used to indirectly evaluate the performance of a structure by comparing it with a limited standard displacement. Therefore, most studies have focused on the estimation of displacement at a certain point, such as the girder center, using other response data. However, if internal forces such as axial force and moment can be estimated, the performance of the structure can be more precisely evaluated by comparing the internal force with the section strength. Based on the stiffness method, the internal force of the structure can be derived using displacement and rotation. The internal forces of each element were determined by using the relative displacement and rotation of the end nodes. The deformed shape was estimated in this study, including the relative displacement at all the nodes of the cable-stayed bridge. Then, the internal forces (girder axial force, girder moment, and cable force) were estimated by applying the deformed shape as a displacement force to the numerical model. The following three cases according to the number of each response data are considered for the estimation of the internal force: case 1 (ω 1, ø 0, ε 0), case 2 (ω 1, ø 6, ε 6), and case 3 (ω 7, ø 6, ε 6). Case 1 represents insufficient data and case 3 represents sufficient data. The results of case 2 were used to validate the effect of the slope and strain data. The displacement measurement location of case 1 is the center of the two-span girder. The measurement locations of case 2 and case 3 are shown in Figure 5c and Figure 6c, respectively.

#### 5.2.1. Girder Axial Force

The girder axial force that significantly occurs owing to the horizontal component of the cable force should be considered to evaluate the performance of cable-stayed bridges. Figure 9 presents examples of the initial, target, and estimated girder axial forces for TM4. The initial line represents the axial force derived from the initial shape analysis, considering the dead load. The maximum axial force is generated at the junction between the girder and pylon, and the minimum axial force is generated at the center of the girder.

Figure 10 presents the normalized absolute percentage error (NAPE) of the axial force along the girder span length for TM4. The estimation error of case 2 dramatically decreases compared to that of case 1, and the estimation error of case 3 is almost zero. These results indicate that slope and strain data can be used as additional data to improve the estimation accuracy, and the exact axial force can be estimated if sufficient multi-response measurement data are provided. The tendency for improvement in the estimation accuracy is represented equally in Table 3. For all cases and TMs, the girder axial force of the cable-stayed bridge can be properly estimated using SRALMR.

#### 5.2.2. Girder Moment

A girder moment is an important factor that should be considered when evaluating the performance of bridges with long spans, such as cable-stayed bridges, where the distribution of girder moment is complex owing to the vertical component of the cable force. An example of the initial, target, and estimated girder moments for TM4 is shown in Figure 11. In the initial shape analysis, the positive moment was mainly large at the girder center, and the negative moment was mainly large at the connection between the girder and pylon. The final girder moment is then determined according to the location and magnitude of the live load.

Figure 12 presents the NAPE of the moment along the girder span length for TM4. Based on the stiffness method, the girder moment was determined by the relative displacement and rotation of the end node at the element. Even a small error in the relative displacement and rotation can cause a large moment-estimation error. SRALMR estimates the deformed shape by minimizing the difference between the ARS and MD only at the locations where the sensors are installed. Additionally, the concentrated load applied to the target model caused a large relative displacement and rotation near the location where the load was applied. Therefore, the estimation error for the moment is generally larger than that for the axial force, and a large error occurs, particularly at the load locations. The maximum NAPEs of the girder moment for TM4 were 80.15, 52.45, and 18.98% for case 1, case 2, and case 3, respectively. The addition of measurement data significantly decreased the estimation accuracy.

The NMAPEs for all cases and TMs are presented in Table 4. The estimated moments for TM1 and TM2 considering one load were more accurate than those for TM3 and TM4 considering multiple loads because the moment distributions of TM1 and TM2 are simpler than those of TM3 and TM4, respectively. In addition, using additional slope and strain data, case 2 can estimate the girder moment more accurately than case 1 for all TMs, and the estimation error of case 3 using a sufficient number of response data is lower than 5%. Therefore, SRALMR can be applied to estimate the girder moment of a cable-stayed bridge if a sufficient amount of response data can be used.

#### 5.2.3. Cable Axial Force

A cable that allows the bridge to have a long span is the main characteristic of a cable-stayed bridge. Owing to this characteristic, the cable suffers a significant tension force and should be considered when evaluating the performance of cable-stayed bridges. The cable force that suffers only from tension can be easily determined using the relative displacement at both ends of the cable. Therefore, the deformation of the girder and pylon should be estimated precisely to determine the proper cable axial force.

Figure 13 represents examples of the initial, target, and estimated cable forces for TM4. The target and estimated cable forces were determined by superposing the cable forces generated by the dead and live loads. Owing to the dead load, the cable forces near both ends of the girder and the girder center are relatively large. The NMAPE for each case for TM4 is shown in Figure 14. A large estimation error of approximately 10% occurred at a certain cable for case 1. However, the estimation error for case 2 dramatically decreases compared to that of case 1, and the estimation error of case 3 is almost zero. Similar to the girder axial force and moment, strain and slope data can be used as additional data to improve the estimation accuracy for the cable force. In addition, the exact cable force can be determined if sufficient multi-response data are provided. The tendency for improvement in the estimation accuracy is represented equally in Table 5.

## 6. Conclusions

In this study, the applicability of SRALMR was extended to the cable-stayed bridge that has complex structural behavior beyond the beam and truss previously validated in [30]. To apply SRALMR to the cable-stayed bridge, an analysis scheme including initial shape analysis was proposed, and performance was verified with four numerical target models. The deformed shape of the cable-stayed bridge was estimated according to the various combinations of limited displacement, slope, and strain data. Then, the girder axial force, girder moment, and cable force were determined for three sensor layout cases using the deformed shape as the displacement force. The numerical validation can be summarized as follows:The deformed shape of the cable-stayed bridge can be well estimated by SRALMR using various combinations of displacement, slope, and strain data. In addition, estimation results show that slope and strain data can enhance the estimation accuracy and reduce the required number of displacement data.From the deformed shape estimated by SRALMR, internal force (girder axial force, girder bending moment, cable axial force) can be properly determined according to the limited amount of response data. A greater amount of used response data enhances the accuracy of internal force estimation.

The results of this study evidence the applicability of SRALMR to the cable-stayed bridges that have complex behavior compared to beams and trusses. However, future studies for validation using experimental data in laboratory and on-site schemes are required to apply SRALMR to the real structure.

## Figures and Tables

**Figure 1 sensors-22-03745-f001:**
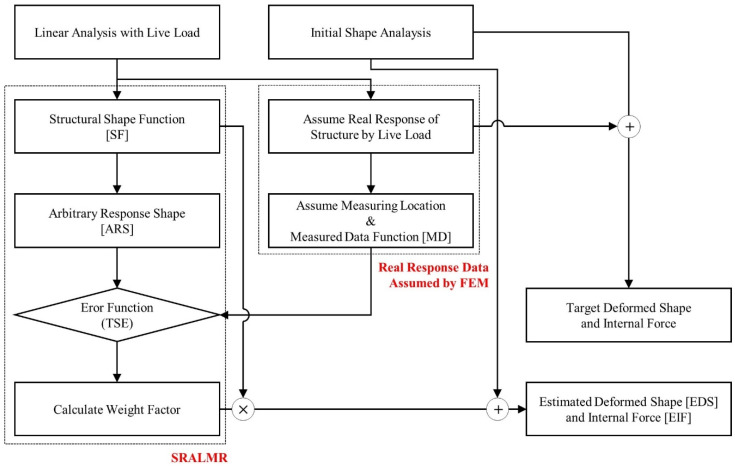
Validation analysis scheme of SRALMR for cable-stayed bridge.

**Figure 2 sensors-22-03745-f002:**
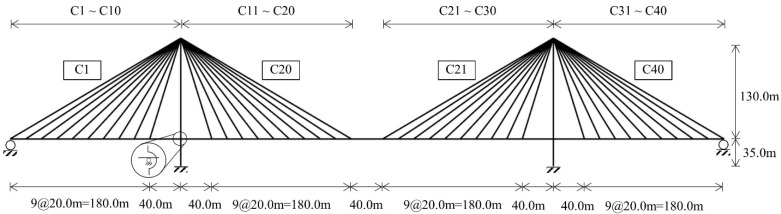
Validation model configuration.

**Figure 3 sensors-22-03745-f003:**
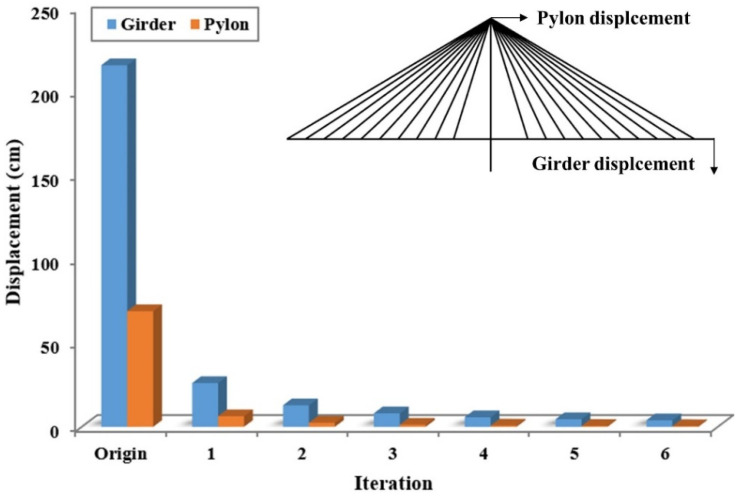
Pylon and girder displacement in each stage of initial shape analysis.

**Figure 4 sensors-22-03745-f004:**
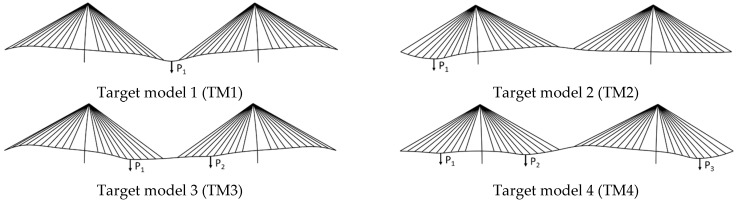
Deformed shape of target model.

**Figure 5 sensors-22-03745-f005:**
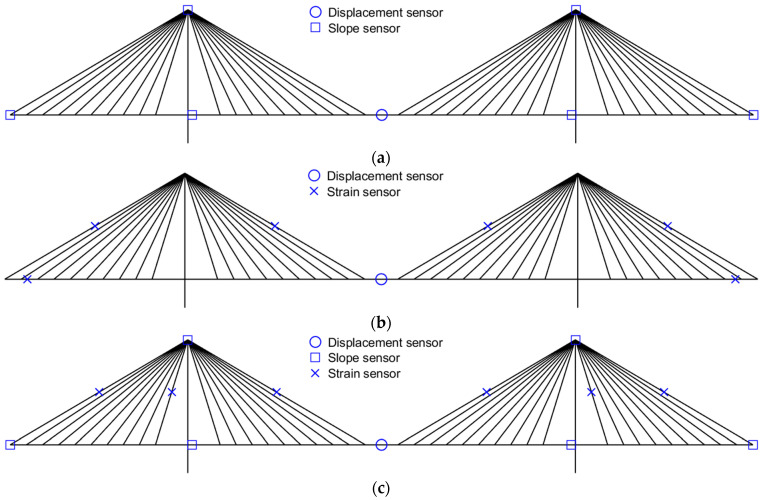
Example of measurement location using 1 displacement sensor. (**a**) *ω* = 1, ø = 6, *ε* = 0; (**b**) *ω* = 1, ø = 0, *ε* = 6; (**c**) *ω* = 1, ø = 6, *ε* = 6.

**Figure 6 sensors-22-03745-f006:**
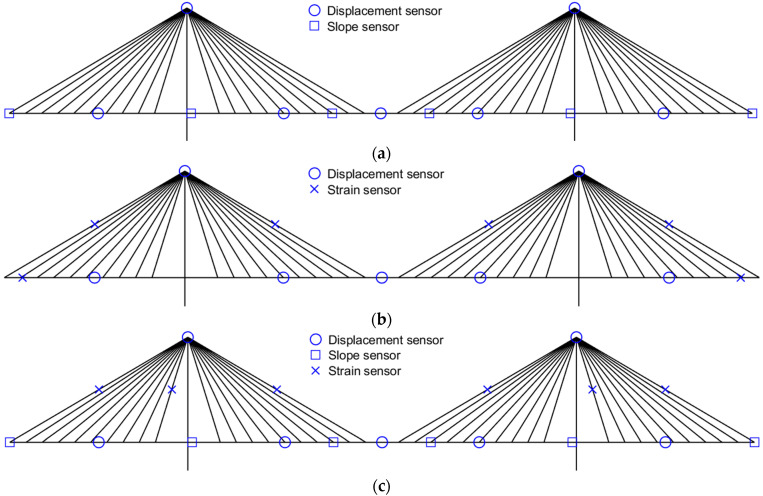
Example of measurement location using 7 displacement sensors. (**a**) *ω* = 7, ø = 6, *ε* = 0; (**b**) *ω* = 7, ø = 0, *ε* = 6; (**c**) *ω* = 7, ø = 6, *ε* = 6.

**Figure 7 sensors-22-03745-f007:**
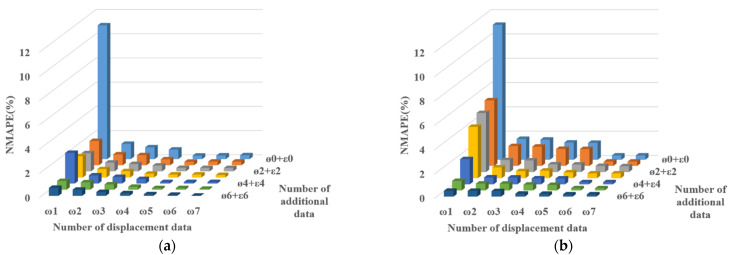
NMAPE of the EDS according to the number of measurement data. (**a**) TM1; (**b**) TM2; (**c**) TM3; (**d**) TM4.

**Figure 8 sensors-22-03745-f008:**
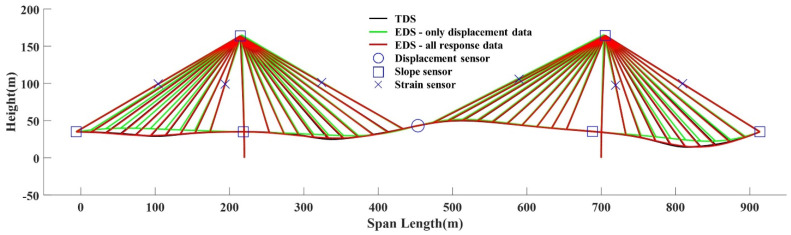
Example of EDS for TM4.

**Figure 9 sensors-22-03745-f009:**
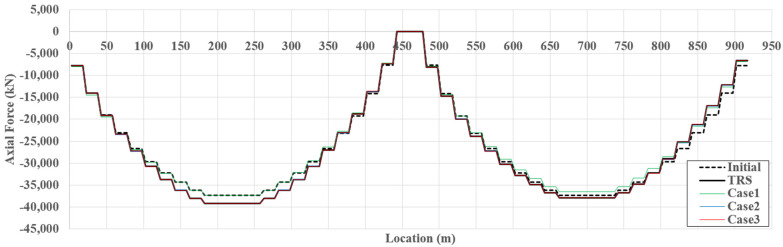
Example of estimated girder axial force for TM4.

**Figure 10 sensors-22-03745-f010:**
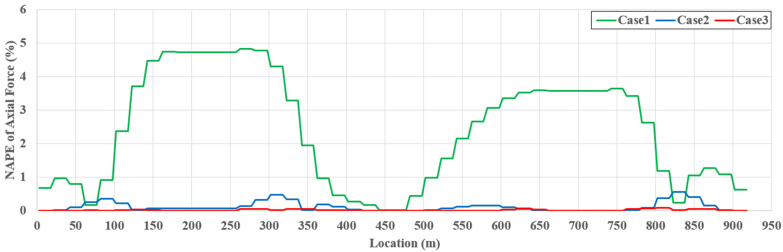
NAPE of estimated girder axial force for TM4.

**Figure 11 sensors-22-03745-f011:**
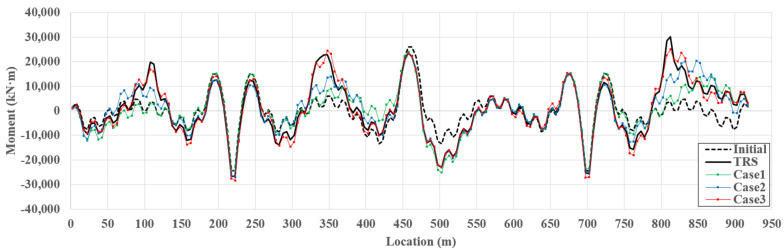
Example of estimated girder moment for TM4.

**Figure 12 sensors-22-03745-f012:**
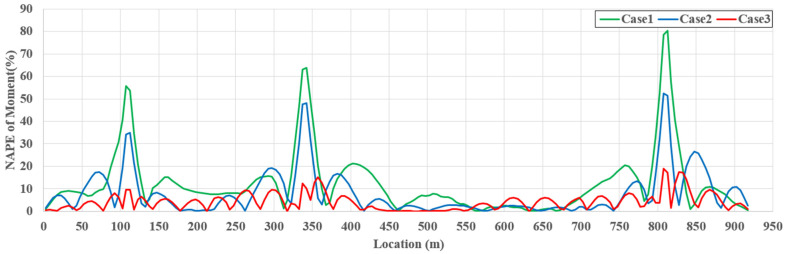
NAPE of estimated girder moment for TM4.

**Figure 13 sensors-22-03745-f013:**
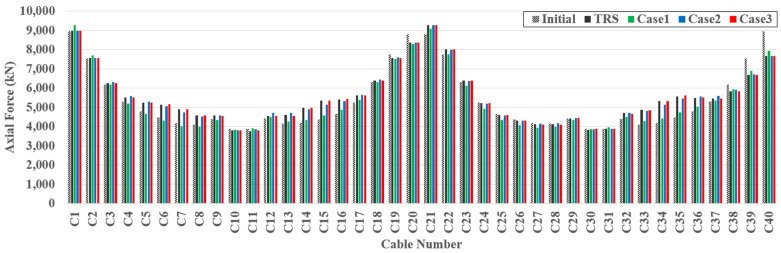
Example of estimated cable axial force for TM4.

**Figure 14 sensors-22-03745-f014:**
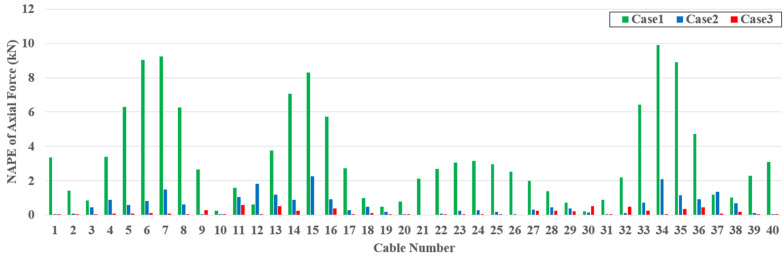
NAPE of estimated cable axial force for TM4.

**Table 1 sensors-22-03745-t001:** Material and geometric properties of main members.

	Girder	Pylon	Cable
Elastic modulus *E* (kN/m^2^)	2.1 × 10^8^	2.1 × 10^8^	2.1 × 10^8^
Sectional *A* (m^2^)	0.749	0.374	0.02
2nd moment of inertia *I* (m^4^)	1.446	3.143	−
Unit weight *γ* (kN/m^3^)	218.27	76.90	76.90

**Table 2 sensors-22-03745-t002:** Details of applied force for target models.

	Force (kN)			
	TM1	TM2	TM3	TM4
P_1_	5000	5000	5000	2000
P_2_	-	-	3000	2500
P_3_	-	-	-	3000

**Table 3 sensors-22-03745-t003:** NMAPE of estimated girder axial force for each TM and case.

	TM1	TM2	TM3	TM4
Case 1	1.471	2.275	2.819	2.382
Case 2	0.332	0.074	0.157	0.115
Case 3	0.002	0.021	0.022	0.019

**Table 4 sensors-22-03745-t004:** NMAPE of estimated girder moment for each TM and case.

	TM1	TM2	TM3	TM4
Case1	6.024	7.127	12.891	12.410
Case2	2.240	3.261	6.647	8.035
Case3	0.345	2.238	3.458	4.164

**Table 5 sensors-22-03745-t005:** NMAPE of estimated cable axial force for each TM and case.

	TM1	TM2	TM3	TM4
Case1	2.905	3.320	3.495	3.406
Case2	0.773	0.335	0.615	0.580
Case3	0.008	0.143	0.132	0.146

## Data Availability

The data presented in this study are available on request from the corresponding author.

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
