# Peer review of "Structural Responses Estimation of Cable-Stayed Bridge from Limited Number of Multi-Response Data"

_sensors, 2022, doi:10.3390/s22103745_

Round 1

Reviewer 1 Report

Structural response analysis using a limited amount of multi-Response (SRALD) data was proposed in this paper and the structural responses of a cable-stayed bridge were analyzed using the proposed approach with the complex structural behavior.  The innovative part of the proposed approach could be emphasized in the introduction section. What is the major difference between SRALD and the traditional structural response analysis approaches?  The industrial application and significance of the proposed approach could be enriched in the conclusion section.

Reviewer 2 Report

The paper entitled “Structural Responses Estimation of Cable-Stayed Bridge from Limited Number of Multi-Response Data” provides a very interesting research work about a proposed methodology for the computation of the health status of cable-stayed bridges, thanks to the outcomes of a number of sensors. The manuscript is well written, easy to read and provides very useful information about innovative applications for Structural Health Monitoring of Bridge structural systems. However, a number of revisions are suggested, in order to increase the value of the manuscript.

  1. Literature review is good for all the covered topics. However, Authors are encouraged to deepen the initial part of the Introduction section, where general Structural Health Monitoring strategies and sensors are described. In addition, special attention has been focused on the bridge deck. Nonetheless, also piers could represent vulnerable elements, as described in the followings; further discussion is needed.

  2. In section 5 validation results are presented. The numerical model of the case study structure has to be much more detailed: which software has been used? How were all the elements modeled (piers, deck, cables)? Linear or non-linear? Some global characteristics are needed.

  3. Graphical results of the EI-DPR-293 distance method should be added, by showing the final configuration of sensors adopted for the validation study.

  4. What are the loading conditions of the case study structural system? Did Authors consider traffic loading, or no dynamic load has been applied? Further description and discussion are needed.

Round 2

Reviewer 2 Report

Authors have significantly improved the quality of the manuscript. Therefore, the article can be accepted for publication in its revised form.

Author Response

Thanks for your comments to improve the quality of the manuscript.

This manuscript is a resubmission of an earlier submission. The following is a list of the peer review reports and author responses from that submission.